# Fetal abdominal obesity in women with one value abnormality on diagnostic test for gestational diabetes mellitus

**Wonjin Kim** [1]☯, **Soo Kyung Park** [2]☯, **Yoo Lee Kim** [1]*

1 Division of Endocrinology and Metabolism, Department of Internal Medicine, CHA Gangnam Medical Center, CHA University School of Medicine, Seoul, Korea, 2 Department of Biostatics and Data Science, School of Public Health, University of Texas Health Science Center at Houston, Houston, Texas, United States of America

☯ These authors contributed equally to this work.
* ylkim@cha.ac.kr, ylkim60@daum.net

**Data Availability Statement:** All relevant data are within the manuscript and its Supporting Information files.

**Funding:** The author(s) received no specific funding for this work.

## Abstract

Previous studies have shown that fetal abdominal obesity (FAO) was already observed at the time of gestational diabetes mellitus (GDM) diagnosis and persisted until delivery despite management in older and/or obese women. In this study, we investigated whether fetuses of women with milder hyperglycemia than GDM have accelerated abdominal growth, leading to adverse pregnancy outcomes. We retrospectively reviewed the medical records of 7,569 singleton pregnant women who were universally screened using a 50-g glucose challenge test (GCT) and underwent a 3-h 100-g oral glucose tolerance test (OGTT) if GCT result was ≥140mg/dL. GDM, one value abnormality (OVA), and normal glucose tolerance (NGT, NGT1: GCT negative, NGT2: GCT positive & OGTT negative) were diagnosed using Carpenter-Coustan criteria. With fetal biometry data measured simultaneously with 50-g GCT, relative fetal abdominal overgrowth was investigated by assessing the fetal abdominal overgrowth ratios (FAORs) of the ultrasonographically estimated gestational age (GA) of abdominal circumference(AC) per actual GA by the last menstruation period(LMP), biparietal diameter(BPD) or femur length(FL), respectively. FAO was defined as FAOR ≥90th percentile The FAORs of GA-AC/GA-LMP and GA-AC/GA-BPD were significantly higher in OVA subjects compared to NGT subjects but not in NGT2 subjects. Although the frequency of FAO in OVA (12.1%) was between that of NGT (9.6%) and GDM (18.3%) without statistically significant difference, the prevalence of large for gestational age at birth and primary cesarean delivery rates were significantly higher in OVA (9.8% and 29.7%) than in NGT (5.1% and 21.5%, p<0.05). Particularly, among OVA subjects with FAO, the prevalence (33.3% and 66.7%) was significantly higher than in those without FAO (9.7% and 24.2%, p<0.05). The degree of fetal abdominal growth acceleration in OVA subjects was intermediate between that of NGT and GDM subjects. OVA subjects with FAO at the time of GDM diagnosis were strongly associated with adverse pregnancy outcomes.

**Competing interests:** The authors have declared that no competing interests exist.

## Introduction

Gestational diabetes mellitus (GDM), defined as glucose intolerance newly detected during pregnancy, is the most common metabolic abnormality encountered in routine obstetric practice, affecting up to 14% of pregnant women. Although various diagnostic methods have been employed, GDM is usually diagnosed through either a one-step 75-g oral glucose tolerance test (OGTT), recommended by the International Association of Diabetes and Pregnancy Study Groups (IADPSG) [1], or a two-step test endorsed by the American College of Obstetricians and Gynecologists (ACOG) which consisted of 3-hour 100-gram OGTT after universal screening with 50-gram glucose challenge test (GCT) performed between 24–28 weeks of gestation [2].

The ACOG diagnostic criteria were originally designed to predict the development of type 2 diabetes mellitus (T2DM) in mothers post-pregnancy [3]. On the other hand, the IADPSG criteria were to predict adverse pregnancy outcomes, use a lower glucose threshold than the ACOG criteria, and diagnose GDM with one abnormal blood glucose value. As a result, the prevalence of GDM doubled compared to that determined by the ACOG criteria [4–6].

It is well established that GDM diagnosed with ≥2 abnormal values by ACOG criteria is associated with increased maternal and perinatal morbidities [3, 7–9]. But controversy remains regarding the clinical risk of an isolated one-value abnormality (OVA) on the OGTT, with mixed results showing no risk [10, 11] or an increased risk of adverse outcomes, such as large for gestational age (LGA) infants [12], macrosomia [13], cesarean section [14], and preeclampsia [15].

Considering the degree of hyperglycemia and its impact on pregnancy outcomes, OVA subjects are very similar to the group of patients who were not diagnosed with GDM by ACOG criteria but were additionally diagnosed by IAPGS criteria. However, the clinical significance of this particular subject group has not been adequately addressed.

Korean Diabetes Association currently recommends using either ACOG or IADPSG tests to diagnose GDM [16], despite significant differences in diagnostic thresholds and resultant pregnancy outcomes [17] between the two criteria. Investigating the clinical significance of OVA in pregnant women is thought to be important for the appropriate management of glucose intolerance during pregnancy.

In our previous retrospective studies, we investigated fetal abdominal overgrowth indicative of abdominal obesity associated with maternal glucose intolerance. In these studies, we observed that fetal abdominal obesity (FAO), detected at 24–28 gestational week (GW) in older and/or obese GDM women [18], was already present at 20–24 GW before the GDM diagnosis [19]. FAO persisted until delivery, even with appropriate treatment of GDM, leading to an increased risk of cesarean section, LGA infants and macrosomia [20]. We also reported recently that older pregnant women with normal glucose tolerance (NGT) showed fetal abdominal overgrowth regardless of maternal obesity [21].

In the present study, we aimed to determine whether women with OVA on the 3-hour 100g OGTT at 24–28 GW had an increased risk of FAO and subsequent adverse pregnancy outcomes, compared with [1] women with a normal GCT (NGT1), [2] women with abnormal GCT but normal OGTT (NGT2), and [3] women with GDM.

## Materials and methods

### Subjects and data collection

We retrospectively reviewed the medical records of 7,820 singleton pregnant women who were followed up and delivered at the CHA Gangnam Medical Center in Seoul, South Korea from January 1, 2012, to April 30, 2015 [18]. Data collection was approved by the Institutional Review Board (IRB) of CHA Gangnam Medical Center, with a waiver of informed consent

granted for the retrospective chart review (IRB No.GCI-18-10). Those data were accessed from May 2018 to March 2019 for research purposes. All methods were performed in compliance with the relevant guidelines and regulations. The Data included information on maternal age during pregnancy, maternal anthropometric measurements, and results of biochemical tests, all of which were obtained from the medical records. Additionally, we collected fetal biometry data measured at the time of the 50-g GCT at 24–28 GW, with the corresponding gestational age (GA) when the ultrasound scans were performed. Body mass index (BMI) was calculated using the formula: BMI = body weight (kg) / (height [m])$^2$.

## Study population

All the pregnant women were universally recommended to undergo screening with a 50-g GCT at 24–28 GW, regardless of fasting status. Subsequently, if the GCT result was ≥140 mg/dL, they proceeded to a 3-h 100-g OGTT with measurements of fasting plasma insulin and HbA1c after fasting for more than 8 hours. GDM, OVA, and NGT were diagnosed using the ACOG (Carpenter-Coustan) criteria. NGT subjects were divided into NGT1 (GCT normal), and NGT2 (GCT abnormal but OGTT normal). GDM subjects received education on proper dietary management, appropriate exercise, and insulin treatment when necessary for glycemic control, as previously described [20]. In contrast, OVA subjects were informed of having one abnormal value on 100-g OGTT but did not undergo additional blood glucose testing.

## Fetal biometry

Of the total 7569, we collected fetal biometry data from 5,562 pregnant women who underwent ultrasound scans on the same day as the 50-g GCT at 24–28 GW. There were no significant differences in the clinical and biochemical characteristics of the groups with and without fetal biometric data. Since the diagnosis of GDM was made within one week after screening, we use "at the time of GDM diagnosis" to describe the fetal biometry performed on the same day of the 50-g GCT and abbreviate this phrase to "at diagnosis." In 87% of these women, gestational dating was confirmed through fetal ultrasonography conducted before 14 weeks of gestation. Biparietal diameter (BPD), femur length (FL), and abdominal circumference (AC) were measured three times using ultrasonography (GE Healthcare, USA) by one of the three experienced sonographers. The mean values were converted to each estimated GA (i.e., GA-BPD, GA-FL, and GA-AC) according to the Japanese fetal growth chart [22, 23].

We calculated fetal abdominal overgrowth ratios (FAORs) as follows: GA-AC/GA-LMP (actual GA measured by the last menstruation period (LMP)) to correct for the variations in ultrasound scan timing, and GA-AC/GA-BPD or GA-AC/GA-FL to detect abdominal overgrowth relative to the head and femur growth, respectively. We defined the presence of FAO at diagnosis of GDM as FAOR ≥90$^{th}$ percentile of the total subjects with fetal biometry (GA-AC/GA-LMP ≥1.080, GA-AC/GA-BPD ≥1.071, GA-AC/GA-FL ≥1.069, respectively). The estimated fetal weight was calculated using the Shinozuka formula [24]. We defined LGA at birth as ≥90$^{th}$ percentile of GA–matched birth weight according to the report of the Committee of the Korean Society of Neonatology by Lee *et al.* [25]. Macrosomia was defined as an infant with a birth weight ≥4 kg.

## Biochemical analysis

Plasma levels of glucose, insulin, and HbA1c were measured using established methods as detailed in a previous study [18]. Insulin resistance (homeostatic model assessment for insulin resistance [HOMA-IR]) and secretion (HOMA-β) were calculated using the homeostasis model assessment method [26].

## Study design

The prevalence of OVA was calculated among the entire 7,569 subjects and subgroups of maternal age and pre-pregnancy BMI: group 1 (age < 35 years and BMI <25 kg/m$^2$ [n = 4,665]), group 2 (age <35 years and BMI ≥25 kg/m$^2$ [n = 262]), group 3 (age ≥35 years and BMI <25 kg/m$^2$ [n = 2,414]), and group 4 (age ≥35 years and BMI ≥25 kg/m$^2$ [n = 228].

We compared clinical and biochemical parameters of pregnant women, fetal biometric data, and pregnancy outcomes among 6,721 NGT (NGT1: 6,171 and NGT2: 550), 248 OVA, and 378 GDM subjects.

In subgroup analysis, we compared the clinical features of OVA subjects according to the presence or absence of FAO at diagnosis, maternal age (≥35 years or <35 years), pre-pregnant BMI (≥25 kg/m$^2$ or <25 kg/m$^2$), and the timing of the abnormal glucose value during the 3-h 100-g OGTT- specifically, at fasting (OVA-0), 1 hour (OVA-1), 2 hours (OVA-2), and 3 hours (OVA-3) after glucose load.

## Statistical analysis

The clinical and biochemical characteristics were summarized as mean and standard deviation. They were compared among subjects with four different glucose tolerances: NGT1, NGT2, OVA, and GDM. Differences across these groups were assessed using either one-way ANOVA or the Kruskal-Wallis test. For pairwise comparison, Tukey's method and Dunnett's test were carried out for one-way ANOVA and Kruskal-Wallis test, respectively (Table 1). Fetal biometry data were also summarized and compared among these groups in the same way as the clinical and biochemical characteristics. For fetal biometry data, only the participants who had valid input for the fetal biometry variables were included (Table 2). Furthermore, using fetal biometry data, the odds of having FAO at GDM diagnosis for NGT2, OVA, and GDM subjects were estimated in comparison to NGT1 subjects conducting univariable logistic regression analysis (Table 3). The prevalence of FAO at diagnosis, 24–28 GW, and pregnancy outcomes among NGT, OVA, and GDM subjects were summarized and compared using the Chi-square test (Table 4).

In addition, subgroup analyses were conducted for OVA subjects, including only those with fetal biometry data (OVA: n = 174, NGT1: n = 4686). Clinical characteristics and pregnancy outcomes were compared by the presence or absence of FAO at diagnosis using the two-sample t-test for clinical characteristics and test of proportions for pregnancy outcomes (Table 5). The prevalence of OVA was compared based on dichotomized pre-pregnancy BMI (≥25 kg/m$^2$, <25 kg/m$^2$) and age (≥35 years, <35 years) using the Chi-square test (S1 Table in S1 File). The prevalence of FAO and pregnancy outcomes in the subjects with OVA according to maternal age and pre-pregnancy BMI were compared with those in NGT1 subjects by test of proportions (S2 Table in S1 File). Furthermore, clinical characteristics and pregnancy outcomes according to the timing of abnormal glucose values on 3-h 100-g OGTT in the OVA subjects were conducted in the same way as analyzing the four groups with different glucose tolerance (S3 Table in S1 File). In the FAO prevalence analysis and subgroup analysis according to the presence or absence of FAO, FAO was defined only with a FAOR of GA-AC/ GA-LMP ≥1.080.

## Results

### Prevalence of OVA according to maternal age and BMI

Of the 7820 initial subjects, 251 were excluded due to pregnancy-induced hypertension before 24 GW (n = 25), no maternal weight records (n = 28), and missing GCT results (n = 198).

**Table 1. Clinical and biochemical characteristics in NGT, OVA, and GDM subjects.**

| | Total NGT (n = 6,721) | NGT | | OVA (n = 248) | GDM (n = 378) | P* |
|---|---|---|---|---|---|---|
| | | NGT1 (n = 6,171) | NGT2 (n = 550) | | | |
| **Clinical** | | | | | | |
| Age (years) | 33.1 ± 3.8 | 33.0 ± 3.8 | 33.7 ± 3.8[a] | 34.4 ± 3.8[a,b] | 35.3 ± 3.9[a,b,c] | <0.0001 |
| Height (cm) | 162.5 ± 4.9 | 162.5 ± 4.9 | 162.2 ± 4.7 | 161.9 ± 5.0 | 161.5 ± 4.8[a] | 0.0001 |
| Prepregnancy BMI (kg/m$^2$) | 20.6 ± 2.6 | 20.6 ± 2.5 | 20.7 ± 2.7 | 21.7 ± 3.5[a,b] | 22.3 ± 3.5[a,b,c] | 0.0001 |
| Weight Gain (kg) | | | | | | |
| Prepregnancy–at diagnosis | 7.5 ± 3.3 | 7.5 ± 3.3 | 7.5 ± 3.2 | 7.6 ± 3.1 | 7.8 ± 3.6 | 0.7951 |
| At diagnosis–near term | 5.4 ± 2.3 | 5.5 ± 2.3 | 5.1 ± 2.3[a] | 4.6 ± 2.6[a,b] | 3.6 ± 2.4[a,b,c] | <0.0001 |
| **Biochemical** | | | | | | |
| 50g-GCT (mg/dL) | 111.7±21.0 | 108.1± 17.6 | 152.8 ± 12.2[a] | 155.4 ± 13.8[a] | 164.9 ± 24.8[a] | <0.0001 |
| 100g-OGTT (mg/dL) | | | | | | |
| fasting | - | - | 80.2 ± 6.4 | 83.1 ± 7.6[b] | 89.2 ± 14.7[b,c] | 0.0001 |
| 1hr | - | - | 140.7 ± 21.2 | 162.7 ± 22.3[b] | 190.6 ± 30.7[b,c] | 0.0001 |
| 2hr | - | - | 126.2 ± 16.7 | 146.6 ± 21.8[b] | 179.3 ± 32.0[b,c] | 0.0001 |
| 3hr | - | - | 110.9 ± 16.1 | 127.3 ± 19.2[b] | 153.5 ± 33.4[b,c] | 0.0001 |
| HbA1c at diagnosis (%) | - | - | 5.0 ± 0.3 | 5.1 ± 0.3[b] | 5.3 ± 0.5[b,c] | 0.0001 |
| (mmol/mol) | | | 31.3 ±3.5 | 32.3±3.5 | 34.4 ± 6.0 | 0.0001 |
| Insulin (ng/dL) | | | | | | |
| fasting | - | - | 8.0 ± 4.2 | 9.5 ± 5.0 | 9.9 ± 5.0[b] | 0.0105 |
| 1hr | - | - | 96.1 ± 107.0 | 85.4 ± 46.5 | 83.9 ± 59.0 | 0.7562 |
| HOMA-IR | - | - | 1.6 ± 0.9 | 2.0 ± 1.1[b] | 2.2 ± 1.2[b] | 0.0001 |
| HOMA-ß | - | - | 181.0 ± 107.6 | 161.9 ± 89.6 | 158.6 ± 99.4 | 0.2174 |

NGT, normal glucose tolerance; OVA, one value abnormality; GDM, gestational diabetes mellitus; BMI, body mass index; GCT, glucose challenge test; OGTT, oral glucose tolerance test; HbA1c, glycated hemoglobin; HOMA-IR, homeostatic model assessment for insulin resistance; HOMA-β, homeostatic model assessment for insulin secretion.

*P-value to an overall comparison among NGT (GCT-), NGT2 (GCT+ OGTT-), OVA and GDM

[a]$P$ <0.05 compared to NGT1

[b]$P$ <0.05 compared to NGT2

[c]$P$ <0.05 compared to OVA.

Consequently, a total of 7,569 subjects who underwent screening with a 50-g GCT were included in this study. Among these 7,569 subjects, 1,233 women had a glucose level of ≥140mg/dL on the 50-g GCT. Out of these, 1,186 (96.2%) underwent a 100-g OGTT, while 47 did not. Among the 1,186 subjects, 552 had no abnormalities, 250 had OVA, and 384 had GDM. This analysis comprises a total of 7,347 subjects which includes 6,171 NGT1, 550 NGT2, 248 OVA, and 378 GDM subjects. A total of 174 subjects (165 NGT1, 2 NGT2, 2 OVA, and 6 GDM) who delivered at other hospitals were excluded from the analysis (S1 Fig in S1 File).

The overall incidence of NGT2, OVA, and GDM was 7.3%, 3.3%, and 5.1%, respectively (Table 1). The prevalence of OVA exhibited significant differences according to maternal age and BMI, with the lowest prevalence of 2.5% in the younger (<35 years) and non-obese (BMI <25 kg/m$^2$) women, while the highest prevalence of 7.5% was observed in the older and obese women (S1 Table in S1 File).

**Table 2. Results of fetal biometry and FAORs measured at diagnosis, 24–28 GW in NGT, OVA, and GDM subjects.**

| | Total NGT (n = 5,097) | NGT | | OVA (n = 174) | GDM (n = 291) | P* |
|---|---|---|---|---|---|---|
| | | NGT1 (n = 4,686) | NGT2 (n = 411) | | | |
| **Fetal biometry** | | | | | | |
| GA-LMP (week) | 26.39 ± 0.96 | 26.39 ± 0.97 | 26.35 ± 0.89 | 26.41 ± 0.93 | 26.32 ± 0.91 | 0.4265 |
| GA-AC (week) | 27.15 ± 1.43 | 27.14 ± 1.43 | 27.20 ± 1.39 | 27.41 ± 1.50 | 27.40 ± 1.56[a, †] | 0.0022 |
| GA-BPD (week) | 26.93 ± 1.54 | 26.92 ± 1.54 | 26.98 ± 1.53 | 26.95 ± 1.58 | 26.86 ± 1.44 | 0.7518 |
| GA- FL (week) | 26.91 ± 1.41 | 26.91 ± 1.42 | 26.96 ± 1.33 | 27.15 ± 1.35[a, †] | 26.87 ± 1.28[c] | 0.1034 |
| EFW(g) | 1019.8 ± 162.9 | 1019.3 ± 163.1 | 1025.1 ± 160.2 | 1032.8 ± 165.0 | 1030.2 ± 162.70 | 0.4453 |
| **FAOR** | | | | | | |
| GA-AC/GA-LMP | 1.028 ± 0.040 | 1.028 ± 0.040 | 1.030 ± 0.038 | 1.035 ± 0.042[†] | 1.041 ± 0.047[a,b,†] | <0.0001 |
| GA-AC/GA-BPD | 1.010 ± 0.047 | 1.010 ± 0.047 | 1.009 ± 0.043 | 1.019 ± 0.044[†] | 1.021 ± 0.047[a,b,†] | 0.0001 |
| GA-AC/GA-FL | 1.010 ± 0.045 | 1.010 ± 0.045 | 1.010 ± 0.041 | 1.010 ± 0.045 | 1.020 ± 0.046[a,b,c,†,] | 0.0016 |

FAOR, fetal abdominal overgrowth ratio; NGT, normal glucose tolerance; OVA, one value abnormality; GDM, gestational diabetes mellitus; GW, gestational weeks; GA-LMP, gestational age by last menstruation period; GA-AC, estimated gestational age by abdominal circumference; GA-BPD, estimated gestational age by biparietal diameter; GA-FL, estimated gestational age by femur length; EFW, estimated fetal weight.

*P-value to an overall comparison among NGT (GCT-), NGT2 (GCT+ OGTT-), OVA and GDM

[a] $P < 0.05$ compared to NGT1

[b] $P < 0.05$ compared to NGT2

[c] $P < 0.05$ compared to OVA

[†] $P < 0.05$ compared to Total NGT (NGT1 and NGT2).

## Clinical and biochemical characteristics of the NGTs, OVA, and GDM subjects

Maternal age and pre-pregnancy BMI showed a sequential increase corresponding to the degree of glucose intolerance, with significantly higher values observed in OVA compared to NGT1 and NGT2, and the highest values recorded in GDM. Although maternal weight gain from pre-pregnancy to the time of diagnosis was similar across the study groups, weight gain from the time of diagnosis to near term was significantly lower in NGT2 (5.1 ± 2.3 kg) compared to NGT1 (5.5 ± 2.3 kg), and OVA (4.6 ± 2.6 kg) exhibited significantly lower weight gain than NGT2 (Table 1).

**Table 3. Odds ratio for FAO at diagnosis, 24–28 GW in NGT2, OVA, and GDM subjects.**

| | Fetal Abdominal Obesity[a] Odds Ratio (95% confidence interval) | | |
|---|---|---|---|
| | GA-AC/GA-LMP[b] ≥ 90th | GA-AC/GA-BPD ≥ 90th | GA-AC/GA-FL ≥ 90th |
| NGT1 | 1 (Ref.) | 1 (Ref.) | 1 (Ref.) |
| NGT2 | 1.143 (0.824, 1.585) | 0.762 (0.522, 1.113) | 0.722 (0.492, 1.060) |
| OVA | 1.308 (0.821, 2.086) | 1.407 (0.898, 2.204) | 1.198 (0.745, 1.929) |
| GDM | 2.140[c] (1.565, 2.928) | 1.744[c] (1.254, 2.424) | 1.603[c] (1.143, 2.247) |

FAO, fetal abdominal obesity; GW, gestational weeks; NGT, normal glucose tolerance; OVA, one value abnormality; GDM, gestational diabetes mellitus; GA-AC, estimated gestational age by abdominal circumference; GA-LMP, gestational age by last menstruation period; GA-BPD, estimated gestational age by biparietal diameter; GA-FL, estimated gestational age by femur length.

[a] FAOR ≥90th percentile defined as GA-AC/GA-LMP ≥1.080, GA-AC/GA-BPD ≥1.071, and GA-AC/GA-FL ≥1.069

[b] Gestational age by LMP at diagnosis

[c] $P < 0.05$ compared with NGT(GCT-).

**Table 4. Prevalence of FAO at diagnosis, 24–28 GW, and pregnancy outcomes in NGT, OVA, and GDM subjects.**

| | Total NGT (n = 6,721) | NGT | | OVA (n = 249) | GDM (n = 378) | P* |
|---|---|---|---|---|---|---|
| | | NGT1 (n = 6,171) | NGT2 (n = 550) | | | |
| FAO at diagnosis (%) | 9.6 | 9.5 | 10.7 | 12.1 | 18.3[a,b] | <0.0001 |
| Primipara (%) | 67.3 | 67.1 | 69.5 | 62.7 | 66.7 | 0.018 |
| LGA (%) | 5.1 | 5.0 | 6.3 | 9.8[a] | 13.6[a,b] | <0.0001 |
| Macrosomia (%) | 1.9 | 1.9 | 2.2 | 3.8 | 4.8[a,b] | 0.003 |
| Infant birth weight (g) | 3,197 ± 421 | 3,195 ± 420 | 3,227 ± 432 | 3,235 ± 465 | 3,226 ± 511[a] | 0.0376 |
| Gestational age at delivery (week) | 39.0 ± 1.5 | 39.0 ± 1.5 | 39.0 ± 1.6 | 38.7 ± 1.6 | 38.4± 1.8 | 0.0587 |
| Male sex of infant (%) | 51.4 | 51.4 | 51.9 | 51.6 | 55.9 | 0.401 |
| Primary cesarean delivery (%) | 21.5 | 21.2 | 25.3[a] | 29.7[a] | 30.2[a] | <0.0001 |

FAO, fetal abdominal obesity; NGT, normal glucose tolerance; OVA, one value abnormality; GDM, gestational diabetes mellitus; LGA, large for gestational age.

*$P$-value to an overall comparison among NGT1 (GCT-), NGT2 (GCT+ OGTT-), OVA and GDM

[a]$P$ <0.05 compared to NGT1

[b]$P$ <0.05 compared to NGT2.

Fasting, 1-hour, 2-hour, and 3-hour plasma glucose levels of the 100-g OGTT, as well as HbA1c in OVA, were significantly higher than in NGT2 and were highest in GDM. While HOMR-IR values for OVA (2.0 ± 1.1) and GDM (2.2 ± 1.2) were significantly higher than NGT2 (1.6 ± 0.9), the trend of lower HOMA-β in these groups did not reach statistical significance (Table 1).

**Table 5. Clinical characteristics and pregnancy outcomes by the presence or absence of FAO at 24–28 GW in OVA subjects.**

| | FAO (–) (n = 153) | FAO (+) (n = 21) | P |
|---|---|---|---|
| **Clinical characteristics** | | | |
| Age (years) | 34.3 ± 3.8 | 35.0 ± 2.6 | 0.4256 |
| Pre-pregnancy BMI (kg/m$^2$) | 21.7 ± 3.5 | 22.1 ± 3.3 | 0.6392 |
| Weight gain (kg) | | | |
| Pre-pregnancy–at diagnosis | 7.7 ± 3.0 | 6.8 ± 3.0 | 0.1821 |
| HbA1c* at diagnosis (%) | 5.1 ± 0.3 | 5.1 ± 0.3 | 0.8845 |
| (mmol/mol) | 32.3 ± 3.5 | 32.4 ± 3.5 | |
| HOMA-IR* | 2.1 ± 1.0 | 1.6 ± 1.0 | 0.0938 |
| HOMA-ß* | 162.9 ± 78.1 | 159.9 ± 78.6 | 0.9117 |
| **Pregnancy outcomes** | | | |
| Primipara (%) | 60.8 | 85.7 | 0.026 |
| Male sex of infant (%) | 51.0 | 80.9 | 0.010 |
| LGA (%) | 9.7 | 33.3 | 0.002 |
| Macrosomia (%) | 4.6 | 9.5 | 0.379 |
| Cesarean delivery (%) | 41.1 | 66.7 | 0.027 |
| Primary cesarean delivery (%) | 24.2 | 66.7 | <0.001 |

FAO, fetal abdominal obesity; GW, gestational week; OVA, one value abnormality; BMI, body mass index; HbA1c, glycated hemoglobin; HOMA-IR, homeostatic model assessment for insulin resistance; HOMA-β, homeostatic model assessment for insulin secretion; LGA, large for gestational age.

* Column Ns for the variable are different from the Ns presented in the table

## FAORs and Odds ratio for FAO in NGTs, OVA, and GDM subjects

The FAOR of each group increased gradually according to maternal glucose intolerance. FAORs of NGT2 were not significantly different from those of NGT1, but FAORs of OVA were significantly higher than those of total NGT (GA-AC/GA-LMP: 1.035 ± 0.042 *vs.* 1.028 ± 0.040, GA-AC/GA-BPD: 1.019 ± 0.044 *vs.* 1.010 ± 0.047, $p$ <0.05). The FAOR of GA-AC/GA-FL in OVA was significantly lower than that in GDM, but other FAORs were comparable with those in GDM subjects (Table 2).

Relative to NGT1, odds ratio for FAO by each FAOR showed a tendency to increase according to maternal glucose intolerance, but there was no statistical significance except for those of GDM (Table 3).

## Prevalence of FAO and pregnancy outcomes of the NGTs, OVA and GDM subjects

The prevalence of FAO, defined as $\geq 90^{th}$ percentile of the FAOR of GA-AC/GA-LMP, was not significantly higher in OVA (12.1%) compared to NGT2 (10.7%) and NGT1 (9.6%). The prevalence of FAO was between that of total NGT (9.6%) and GDM (18.3%), without a statistically significant difference (Table 4).

Although infant birth weights were comparable between the study groups, the rate of LGA in OVA (9.8%) was significantly higher than NGT1 (5.4%) and the highest in GDM (13.6%). The frequency of macrosomia in OVA was not significantly higher than in NGT1 (3.8% vs. 2.1%), but it was significantly higher in GDM (4.8%) than in NGT1. Even though NGT2 subjects did not show a higher frequency of LGA and macrosomia than NGT1 subjects, the primary cesarean section rate was significantly higher as 25.3%, along with OVA (29.7%) and GDM (30.2%) (Table 4).

## Clinical features and pregnancy outcomes of OVA subjects according to the presence or absence of FAO

Compared to OVA subjects without FAO at diagnosis, subjects with FAO had a significantly higher frequency of male babies (80.9% vs. 51.0%), LGA at birth (33.3% vs. 9.7%), and primary cesarean delivery (66.7% vs. 24.2%). However, the frequency of macrosomia did not differ between the two groups (Table 5). For the OVA group with FAO, the number of cases was too small, despite the p-value being less than 0.05 in the proportion test for pregnancy outcome.

## Prevalence of FAO and pregnancy outcomes in OVA according to maternal age and BMI

While the prevalence of FAO was significantly higher in the older and obese group 4 compared with NGT1 (27.3% *vs.* 9.5%, $p$ <0.05), LGA at birth and macrosomia were more frequent in the young and obese group 2, and primary cesarean delivery rate was higher in the older and non-obese group 3 (S2 Table in S1 File).

## Clinical feature and pregnancy outcomes of OVA subjects according to the timing of abnormal glucose value in 3-h 100-g OGTT

In OVA-0 subjects showing fasting hyperglycemia, pre-pregnancy BMI was significantly higher than in the other groups and HOMA-IR was significantly higher than in OVA-2 subjects. Additionally, HbA1c level, the frequency of FAO at diagnosis, and primary cesarean delivery rate were higher than the other three groups but did not reach statistical significance (S3 Table in S1 File).

## Discussion

Mild glucose intolerance has been reported to be associated with adverse pregnancy outcomes, including increased incidence of LGA infants, macrosomia, and cesarean delivery [12–15, 27]. Based on our previous reports that FAO was detected already at 24–28 GW in old and/or obese GDM subjects [18], we investigated whether FAO is affected in subjects with one abnormal value on 100-g OGTT (OVA) and in subjects with abnormal GCT but normal OGTT results (NGT2). Fetal abdominal growth was accelerated in OVA subjects but not in NGT2 subjects. The degree of fetal abdominal growth acceleration and FAO frequency in OVA subjects were intermediate between those in NGT and GDM subjects. However, the prevalence of LGA at birth and primary cesarean delivery rate in OVA subjects were significantly higher than those in NGT subjects and somewhat similar to those in GDM subjects. Additionally, OVA subjects with FAO at the time of GDM diagnosis were found to have much more higher LGA prevalence and cesarean section rate.

In this study, our subjects showed a gradual increase in maternal age and pre-pregnancy BMI depending on maternal glucose intolerance. As reported by Di Cianni et al. [28], pre-pregnancy BMI was higher in both OVA and GDM compared to NGT, but maternal age in OVA subjects was between NGT and GDM subjects and was significantly different from each other. The maternal age of NGT2 was significantly higher than in NGT1, while pre-pregnancy BMIs were not significantly different between the two NGT groups.

Even though plasma glucose levels on 50-g GCT were not significantly different among NGT2, OVA, and GDM groups, plasma glucose levels on 100-g OGTT and HbA1c exhibited significant differences among the three groups. Those values in OVA subjects were all between those of NGT2 and GDM subjects. The HOMA-IR of OVA and GDM subjects was significantly higher than of NGT2 subjects and there was no significant difference between them, but HOMA-β decreased gradually in the order of NGT2, OVA, and GDM without statistical significance.

The prevalence of OVA and NGT2 in this study was 3.3% and 7.3%, respectively, which is consistent with a previous study by Forest et.al. [10]. In subgroup analysis, the prevalence of OVA was significantly higher in older and obese mothers compared to younger and non-obese mothers (7.5% vs. 2.5%), although the difference was not as large as the prevalence of GDM (22.4% vs. 3.2%) [18]. The highest prevalence of GDM [18] and OVA in older and obese women was associated with higher HOMA-IR and lower HOMA-β compared to younger and non-obese women, but the decrease in HOMA-β did not reach statistical significance, in OVA subjects

The above clinical and biochemical findings indicate that OVA represents a distinct entity from NGT2 or NGT1 and fall between NGT and GDM or is rather similar to GDM. OVA subjects have impaired insulin secretion and increased insulin resistance associated with maternal age and obesity, as in GDM [18–20, 29], but these defects are milder in OVA subjects [28, 30, 31].

Schaefer-Graf et al. [27] reported that 16.5% of fetuses in OVA and GDM subjects showed FAO indicated by AC ≥90[th] percentile at 24–28 GW. AC measurement of AC by fetal biometry has been used as a reliable marker of FAO [32]. In this study, FAOR was calculated as an indicator of fetal abdominal overgrowth relative to head and femur growth using estimated GAs instead of the percentile values of fetal parts such as AC measured by ultrasound. Our fetal biometric data showed a gradual acceleration of fetal abdominal growth depending on the degree of maternal glucose intolerance, as observed in the HAPO study [33, 34]. Fetal abdominal overgrowth, represented by FAOR, was detected in OVA subjects compared to NGT subjects, but not in NGT2 subjects compared to NGT1 subjects. FAORs of OVA subjects, as well as those of GDM, were significantly higher than those of NGT subjects, but FAORs of NGT2

were not higher than those of NGT1. However, the prevalence of FAO, defined as ≥90th percentile of FAOR, in OVA subjects (12.1%) was not significantly higher than that in NGT subjects (9.6%), whereas it was significantly higher in GDM (18.3%). Our findings suggest that abdominal growth is accelerated in OVA fetuses compared to fetuses of NGT subjects, but not to the same extent as in GDM. Additionally, fetal abdominal growth is not accelerated in NGT2 subjects compared to NGT1 subjects as if clinical characteristics were not different.

Although gestational age at birth and infant birth weight for total NGT, OVA, and GDM subjects were not significantly different from each other, the prevalence of LGA in OVA subjects was significantly higher than in NGT and comparable to GDM. These findings were consistent with previous reports that the frequency of LGA and macrosomia in untreated OVA was even higher than in treated GDM, despite no significant difference in the average birth weight among the three groups [12, 27, 35–37]. The primary cesarean delivery rate of OVA subjects was also significantly higher than that of NGT and comparable to that of GDM, as in other reports [34, 38]. These findings support the existence of an association between mild glucose intolerance and fetall abdominal overgrowth leading to LGA infant, macrosomia, and primary cesarean delivery.

Previous [18] and present data show that GDM and OVA subjects share many clinical characteristics and pregnancy outcomes. Notably, like GDM subjects, OVA subjects with FAO at the time of diagnosis of GDM had a three-fold higher frequency of LGA at birth and more than two-fold higher rates of macrosomia and primary cesarean delivery compared to OVA subjects without FAO. These findings may be explained by the fetal glucose steal. Because the fetal endocrine pancreas already acquires an insulin response to fetal glycemia at 14–20 GW, fetal abdominal overgrowth risk appears early in pregnancy for women developing gestational glucose intolerance [32], especially for women with obesity and/or advanced maternal age. Established fetal hyperinsulinemia well before GDM screening predisposes to fetal abdominal overgrowth and persistently high glucose delivery to the fetus may lower maternal glucose levels, resulting in false negative results in OGTT [39].

In subgroup analysis according to maternal age (35 years) and pre-pregnancy BMI (25 kg/m²), the prevalence of OVA was influenced more by maternal pre-pregnancy BMI than by maternal age as in GDM [18]. Also, LGA at birth was more frequent in obese OVA subjects that in non-obese OVA subjects. These findings were concordant with the idea that maternal obesity has a more pronounced impact on the diagnosis and fetal complications of GDM and OVA than advanced maternal age [40, 41], although the effect may vary depending on the degree of maternal obesity and maternal age distribution

On the other hand, the frequency of FAO in the young and obese group was not higher than in NGT1 but the FAO frequency in the older and obese group was 3-fold higher than in NGT1. But their higher FAO frequency did not lead to a higher frequency of LGA and macrosomia. These findings could be cautiously explained by the relatively well-preserved insulin secretory capacity in young and obese subjects compared to older and obese subjects and fetal growth restriction in advanced maternal age [42] and suggest that there may be differences in fetal growth trajectories between young and older obese women. However, the number of cases in each group was thought to be too small to obtain reliable results.

The timing of abnormal blood glucose levels in the OGTT [28, 43], along with the number of blood glucose levels exceeding the normal range [44], is also an important factor influencing pregnancy outcomes. In a retrospective cohort study including 152 OVA subjects diagnosed by ACOG criteria, those with abnormal fasting glucose had a higher prevalence of macrosomia and primary cesarean delivery [43]. Our OVA subjects with fasting hyperglycemia showed significantly higher pre-pregnancy BMI and HOMA-IR, and their prevalence of FAO at the diagnosis of GDM and primary cesarean delivery rate tend to be higher than the other OVA subjects.

The Frequency of FAO, LGA, and macrosomia in our NGT2 subjects was not different from those of NGT1 and was in between NGT1 and OVA, even though there are reports of association with a higher frequency of LGA and macrosomia [37, 45–47]. However, the primary cesarean delivery rate was significantly higher than in NGT1 in this study. Therefore, this group of subjects also needs concern when they have risk factors for fetal overgrowth, such as maternal obesity and/or advanced age.

As mentioned in the introduction, subjects with OVA are very similar to subjects who were not diagnosed with GDM by ACOG criteria but were additionally diagnosed by IADPSG criteria [17]. The mean plasma glucose values on 2-hr 75-g OGTT for these subjects were very similar to those of OVA in this study. Furthermore, the prevalence of GDM by IADPSG criteria (8.1%) and by two-step test with ACOG criteria (5.6%), observed in a randomized controlled trial [48], were comparable with the combined prevalence of OVA and GDM (8.4%) and GDM alone (5.1%) in our study population. In addition, maternal old age, pre-pregnancy obesity, fasting hyperglycemia, and FAO at the time of GDM diagnosis were associated with adverse pregnancy outcomes in this study. Treating OVA subjects with risk factors observed in this study, including advanced maternal age, pre-pregnancy obesity, fasting hyperglycemia, and FAO at the time of GDM diagnosis, may improve pregnancy outcomes without increasing unnecessary medical burden. Further study is thought to be necessary for this matter.

The limitations of the present study include the single-center, retrospective, and non-interventional observational study design. The number of OVA subjects was too small for subgroup analysis. Above all, inter-observer variability in the assessment *via* ultrasonography was not evaluated due to the retrospective nature of the study. However, there were no differences in the ultrasound scanners used, and all pregnant women scanned were randomly assigned to one of the three sonographers. Nevertheless, the strength of our study is that all subjects were Asian and were managed according to the same clinical protocol during the study period. However, generalization of our findings to other populations may be limited.

In summary, the prevalence of OVA was significantly higher in older and obese women compared to younger and non-obese women as in GDM. Fetal abdominal growth of OVA subjects accelerated at the time of GDM diagnosis, and the extent was between that of NGT and GDM, but fetuses of NGT2 subjects did not show acceleration compared to NGT1 subjects. In OVA subjects, FAO detected at 24–28 GW was associated with increased rates of LGA at birth and primary cesarean delivery, in comparison with subjects without FAO. Advanced maternal age, obesity, and subjects with high fasting blood glucose levels were also associated with adverse pregnancy outcomes. These findings suggest that detection and appropriate treatment may be necessary to prevent FAO and adverse pregnancy outcomes in women with OVA, especially in high-risk groups.

## Supporting information

**S1 File.**
(DOCX)

**S1 Dataset.**
(ZIP)

## Acknowledgments

The authors would like to express sincere gratitude to Prof. WS Park of Gangnam CHA Medical Center, CHA University, for his advice on writing the manuscript.

## Author Contributions

**Data curation:** Wonjin Kim, Yoo Lee Kim.

**Formal analysis:** Soo Kyung Park.

**Investigation:** Wonjin Kim, Soo Kyung Park.

**Methodology:** Soo Kyung Park, Yoo Lee Kim.

**Software:** Soo Kyung Park.

**Supervision:** Yoo Lee Kim.

**Validation:** Soo Kyung Park, Yoo Lee Kim.

**Writing – original draft:** Wonjin Kim, Soo Kyung Park, Yoo Lee Kim.

**Writing – review & editing:** Wonjin Kim, Yoo Lee Kim.

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
