## [Decision Letter · Decision Letter 0]

27 Feb 2024

PONE-D-23-38854Fetal Abdominal Obesity in Women with One Value Abnormality on Diagnostic Test for Gestational Diabetes MellitusPLOS ONE

Dear Dr. Kim,

Thank you for submitting your manuscript to PLOS ONE. After careful consideration, we feel that it has merit but does not fully meet PLOS ONE’s publication criteria as it currently stands. Therefore, we invite you to submit a revised version of the manuscript that addresses the points raised during the review process.

This is an interesting manuscript. There are two well thought out reviews. Both reviewers have concerns about the manuscript, which I agree with, and have suggested revisions. As there are lots of them I do not propose to add anymore. I think that all the points raised by the reviewers need dealing with robustly in any revised manuscript, but in particular reviewer 1's comment about the Discussion section and reviewer 2's points 1, 7 and 8 need dealing with particularly carefully to get this manuscript accepted for publication.==============================

We look forward to receiving your revised manuscript.

Kind regards,

Clive J. Petry, PhD

Academic Editor

PLOS ONE

Reviewers' comments:

Reviewer's Responses to Questions

**Comments to the Author**

1. Is the manuscript technically sound, and do the data support the conclusions?

Reviewer #1: Partly

Reviewer #2: No

2. Has the statistical analysis been performed appropriately and rigorously? 

Reviewer #1: Yes

Reviewer #2: No

3. Have the authors made all data underlying the findings in their manuscript fully available?

Reviewer #1: Yes

Reviewer #2: Yes

4. Is the manuscript presented in an intelligible fashion and written in standard English?

Reviewer #1: No

Reviewer #2: Yes

5. Review Comments to the Author

Reviewer #1: The authors conducted a retrospective investigation into the risk of fetal abdominal obesity in pregnant Korean women, revealing a propensity for fetal abdominal overgrowth in those who tested negative for gestational diabetes mellitus (GDM) according to Carpenter-Coustan criteria but exhibited one abnormal glucose value (OVA) during the 100-g OGTT. Building upon their prior works within the same population, the authors previously focused on GDM and normal glucose tolerant (NGT) women, stratified based on maternal age and pre-pregnancy body mass index.

The manuscript's writing and English are acceptable, and the statistical analyses performed are correct. However, it lacks novelty, as numerous studies have explored the adverse pregnancy outcomes associated with GDM and intermediate glucose intolerance states, following Carpenter-Coustan criteria, in even more contemporary populations.

A key strength of this work, which could be emphasized for the reader, is its focus on a population of Korean women who were not adequately represented in the HAPO study. This gap in representation makes the risk of fetal overgrowth related to GDM as per IADPSG criteria in Korean women less clear. As noted by the Authors themselves, the pregnancy risks of GDM diagnosed according to IADPSG criteria could be similar to OVA as per Carpenter-Coustan glucose cut-offs.

To enhance the manuscript, I recommend a critical refinement of the discussion section, transcending mere result reporting. Fetal abdominal overgrowth risk appears early in pregnancy for women developing gestational glucose intolerance, preceding the typical 24-28 gestational weeks’ window for performing an OGTT. Notably, based on presented data and shared characteristics of GDM and OVA women testing positive for fetal abdominal overgrowth (e.g., with obesity and/or advanced maternal age), it seems that some GDM diagnoses in this study might have been overlooked. This oversight could be attributed to excessive glucose shunting to a large-growing fetus, causing potential "false" negative results during the GCT/100-g OGTT.

Furthermore, I suggest interpreting results in light of recent reports (i.e., doi:10.3390/jcm12082830; doi: 10.1016/j.coph.2021.06.003. ), providing additional insights into observed phenomena and reinforcing the idea that maternal obesity has a more pronounced impact on the diagnosis and fetal complications of GDM and OVA than advanced maternal age (refer to Table S1 and S2).

I believe addressing these points will significantly enhance the manuscript's contribution and relevance. I appreciate your attention to these suggestions and look forward to the improved version of the manuscript.

Reviewer #2: The authors present the results of a single center retrospective cohort study that aimed to examine whether women with the EXPOSURE of a single abnormal value on the 100 gram OGTT at 24-28 GW had an increased risk of the OUTCOMES of 1) fetal abdominal adiposity (FAO) and 2) adverse pregnancy outcomes (APO) when COMPARED with [1] women with a normal GCT (NGT1), [2] women with abnormal GCT but normal OGTT (NGT2), and [3] women with GDM. The subject matter has been assessed previously in many different ways and it is thus well known that glucose tolerance in pregnancy occurs along a continuum with no specific identifiable inflection point. This study adds to this body of evidence but does not excel in its novelty. The authors have tried to do a lot with their datasaet, and perhaps too much, leading to a sometimes confusing and less focused presentation of the results.

The study is well written but there is still some need for editing due to typos and grammatical errors. I have the following comments:

Introduction:

1. While summarizing well some of the screening and diagnostic controversies surrounding GDM I did not get a good idea as a reader as to why the authors chose to focus on the intermediate outcome of FAO as opposed to the actual perinatal outcomes (which are also reported).

2. Line 56 – do you mean > or ≥ “2 abnormal values”

3. Line 60 “Toxemia…” term is not really used anymore

4. Line 73 : should be “older pregnant WOMEN (or people)…”

Methods:

5. Line 105 – 113 : This information would usually be presented in the results as Fig 1 – derivation of the cohort. See STROBE guidelines

6. Was there not a 50 gram GCT result that was diagnostic of GDM (>11.0)? What group were these women assigned to or were they excluded?

7. How do you know that women with 1 abnormal value were not treated in a similar fashion to those with two abnormal values as the results were not blinded to the clinician or patient? The fact that OVA and GDM had the same rate of CS leads me to believe that the OVA group was managed differently than the NGT group.

8. Why do some women have a biometry scan at the time of the GCT and why do some not? Could it be because of additional risk factors? If so, this would bias the results and limit generalizability. If information could be provided regarding the characteristics of those that had a scan and those that dd not could help identify any baseline differences.

9. I have to admit that I don’t exactly understand the creation of the GA-AC. I assume you converted the measured AC (mm) to the corresponding mean AC (mm) by gestational AC nomograms. First, giving an example would make it clearer. Second, I am not understanding why you would not just report percentiles for the AC instead.

10. How was the FAOR ≥90th percentile determined? Was the 90th centile determined internally from the cohort?

11. Please discuss the statistical power of the study for the outcome of FAO and APO. I doubt that the study was powered for significant adverse outcomes.

12. I see no explanation regarding how the adverse perinatal outcomes were chosen as it seems quite arbitrary. I see no elements of severe maternal or neonatal morbidity which are of course the most significant. Were there cases of stillbirth after GDM screening?

13. Just a general comment regarding the use of “adiposity”. You are not measuring adiposity by measuring the AC. This is a sum of fetal liver size, perhaps intraabdominal adiposity and subcutaneous fat. The liver size likely contributes the most to the AC measurement thus the term is inaccurate)

Results:

14. Well controlled GDM should lead to near normalization of the fetal growth and this does not appear to be the case here. Could it be that your GDM population is suboptimally managed and this is the reason for some of these differences? Providing information on adequacy of glycemic control and/or further stratifying by method of control (medical\\lifestyle alone) would be helpful in interpreting the results.

15. Line 265: should be “Clinical features…”

16. For analysis in Table 3 – I would expect to see an adjusted analysis, adjusting at least for age, BMI.

Discussion:

17. “Management of mild GDM reduced the rates of LGA and macrosomia in newborns…” – These are not results from this study. Start the discussion by summarizing the main findings.

18. Too long

19. Line 400: Do not use the term “maternal old age”!

20. Line 402: “Treating only additionally diagnosed GDM by IADPSG criteria with these risk factors may improve pregnancy outcomes without increasing unnecessary medical burden.” – this is totally unsubstantiated and there is evidence that this not true.

21. Limitation: The “all Asian” ethnicity of the cohort makes these results NOT generalizable to other populations

22. Line 420: The results of this study are not able to shed any light on how strict glycemic control would affect any outcomes. This is overinterpretation.

6. PLOS authors have the option to publish the peer review history of their article (what does this mean?). If published, this will include your full peer review and any attached files.

Reviewer #1: No

Reviewer #2: No

---

## [Author Response · Author response to Decision Letter 0]

22 Apr 2024

Response to Reviewers

Note to reviewers and editors: 

We would like to express our gratitude to the reviewers for their constructive comments. In light of their comments, we have revised the manuscript to enhance its weaker sections. It is important to note that we have incorporated the reviewers’ suggestions into the manuscript. All modifications and their respective locations have been documented using tracked changes and revised manuscripts, as per journal policy.

Reviewer #1: The authors conducted a retrospective investigation into the risk of fetal abdominal obesity in pregnant Korean women, revealing a propensity for fetal abdominal overgrowth in those who tested negative for gestational diabetes mellitus (GDM) according to Carpenter-Coustan criteria but exhibited one abnormal glucose value (OVA) during the 100-g OGTT. Building upon their prior works within the same population, the authors previously focused on GDM and normal glucose tolerant (NGT) women, stratified based on maternal age and pre-pregnancy body mass index.

The manuscript's writing and English are acceptable, and the statistical analyses performed are correct. However, it lacks novelty, as numerous studies have explored the adverse pregnancy outcomes associated with GDM and intermediate glucose intolerance states, following Carpenter-Coustan criteria, in even more contemporary populations.

A key strength of this work, which could be emphasized for the reader, is its focus on a population of Korean women who were not adequately represented in the HAPO study. This gap in representation makes the risk of fetal overgrowth related to GDM as per IADPSG criteria in Korean women less clear. As noted by the Authors themselves, the pregnancy risks of GDM diagnosed according to IADPSG criteria could be similar to OVA as per Carpenter-Coustan glucose cut-offs.

To enhance the manuscript, I recommend a critical refinement of the discussion section, transcending mere result reporting. Fetal abdominal overgrowth risk appears early in pregnancy for women developing gestational glucose intolerance, preceding the typical 24-28 gestational weeks’ window for performing an OGTT. Notably, based on presented data and shared characteristics of GDM and OVA women testing positive for fetal abdominal overgrowth (e.g., with obesity and/or advanced maternal age), it seems that some GDM diagnoses in this study might have been overlooked. This oversight could be attributed to excessive glucose shunting to a large-growing fetus, causing potential "false" negative results during the GCT/100-g OGTT.

Furthermore, I suggest interpreting results in light of recent reports (i.e., doi:10.3390/jcm12082830; doi: 10.1016/j.coph.2021.06.003. ), providing additional insights into observed phenomena and reinforcing the idea that maternal obesity has a more pronounced impact on the diagnosis and fetal complications of GDM and OVA than advanced maternal age (refer to Table S1 and S2).

I believe addressing these points will significantly enhance the manuscript's contribution and relevance. I appreciate your attention to these suggestions and look forward to the improved version of the manuscript.

We appreciate the comments that open up new perspectives and encourage us. We have revised the discussion to reflect your comments as much as possible. 

“Previous[18] and present data show that GDM and OVA subjects share many clinical characteristics and pregnancy outcomes. Notably, like GDM subjects, OVA subjects with FAO at the time of diagnosis of GDM had a three-fold higher frequency of LGA at birth and more than two-fold higher rates of macrosomia and primary cesarean delivery compared to OVA subjects without FAO. However, there were no significant clinical features in the subjects showing FAO, even though maternal age and pre-pregnancy BMI were slightly higher in this group. These findings may be explained by the fetal glucose steal. Because the fetal endocrine pancreas already acquires an insulin response to fetal glycemia at 14-20 GW, fetal abdominal overgrowth risk appears early in pregnancy for women developing gestational glucose intolerance[32], especially for women with obesity and/or advanced maternal age. Established fetal hyperinsulinemia well before GDM screening predisposes to fetal abdominal overgrowth and persistently high glucose delivery to the fetus may lower maternal glucose levels, resulting in false negative results in OGTT[39].” (Line 397-409 in the “Revised Manuscript with Track changes” and lines 359-368 in the “Manuscript” file)

“In subgroup analysis according to maternal age (35 years) and pre-pregnancy BMI (25 kg/m2), the prevalence of OVA was influenced more by maternal pre-pregnancy BMI than by maternal age as in GDM[18]. Also, the frequency of LGA at birth was more frequent in obese OVA subjects than in non-obese OVA subjects. These findings were concordant with the idea that maternal obesity has a more pronounced impact on the diagnosis and fetal complications of GDM and OVA than advanced maternal age[40,41], although the effect may vary depending on the degree of maternal obesity and maternal age distribution” (Line 410-418 in the “Revised Manuscript with Track changes” and lines 369-375 in the “Manuscript” file)

Reviewer #2: The authors present the results of a single center retrospective cohort study that aimed to examine whether women with the EXPOSURE of a single abnormal value on the 100 gram OGTT at 24-28 GW had an increased risk of the OUTCOMES of 1) fetal abdominal adiposity (FAO) and 2) adverse pregnancy outcomes (APO) when COMPARED with [1] women with a normal GCT (NGT1), [2] women with abnormal GCT but normal OGTT (NGT2), and [3] women with GDM. The subject matter has been assessed previously in many different ways and it is thus well known that glucose tolerance in pregnancy occurs along a continuum with no specific identifiable inflection point. This study adds to this body of evidence but does not excel in its novelty. The authors have tried to do a lot with their data set, and perhaps too much, leading to a sometimes confusing and less focused presentation of the results.

The study is well written but there is still some need for editing due to typos and grammatical errors. I have the following comments:

Introduction:

1. While summarizing well some of the screening and diagnostic controversies surrounding GDM I did not get a good idea as a reader as to why the authors chose to focus on the intermediate outcome of FAO as opposed to the actual perinatal outcomes (which are also reported).

We previously performed retrospective studies of maternal glucose intolerance and fetal growth. Because the main effect of maternal glucose intolerance on the fetus is increased adiposity, these studies focused on fetal abdominal overgrowth and obesity compared to head and femur growth. Additionally, other reports have shown that fetal abdominal overgrowth is a much more sensitive early indicator of the effects of maternal hyperglycemia than simply LGA or macrosomia. For this reason, we selected FAO as the intermediate outcome for other perinatal outcomes.

We inserted the following sentence in lines 71-72 in the “Revised Manuscript with Track changes” and lines 70-71 in the “Manuscript” file.

“In our previous retrospective studies, we investigated fetal abdominal overgrowth indicative of abdominal obesity associated with maternal glucose intolerance.”

2. Line 56 – do you mean> or ≥ “2 abnormal values”

Thank you very much for pointing out the error. We have corrected the error as you pointed out. (Line 56 in the “Revised Manuscript with Track changes” and in the “Manuscript” file)

3. Line 60 “Toxemia…” term is not really used anymore

Thank you very much for the correction. We revised the term “toxemia” to “preeclampsia.” (Lines 60-61 in the “Revised Manuscript with Track changes” and line 60 in the “Manuscript” file)

4. Line 73: should be “older pregnant WOMEN (or people)…”

Thank you for pointing out the missing word. The word “women” was inserted. (Line 76 in the “Revised Manuscript with Track changes” and line 68 in the “Manuscript” file)

Methods:

5. Line 105 – 113: This information would usually be presented in the results as Fig 1 – derivation of the cohort. See STROBE guidelines

Based on your recommendation, we moved that paragraph to the first part of the results. (Line 180-188 in the “Revised Manuscript with Track changes” and lines 178-186 in the “Manuscript” file)

6. Was there not a 50 gram GCT result that was diagnostic of GDM (>11.0)? What group were these women assigned to or were they excluded?

No subjects were excluded due to the 50-g GCT results. Among 30 patients with a level of 200mg/dL or higher, 3 had NGT, 3 had OVA, and 24 had GDM. 

7. How do you know that women with 1 abnormal value were not treated in a similar fashion to those with two abnormal values as the results were not blinded to the clinician or patient? The fact that OVA and GDM had the same rate of CS leads me to believe that the OVA group was managed differently than the NGT group.

Thank you for pointing out this very important issue. In fact, our intention in writing this paper was to inform our readers that OVA is not normal for pregnant women, since most obstetricians and endocrinologists did not pay attention to pregnant women with OVA at that time, and some of them still do the same. However, as you rightly point out, since the results were not blinded to the clinicians or patients, it is possible that the OVA group was managed differently by obstetricians compared to the NGT group. If that is true, since GDM is a higher risk, shouldn’t the C/S rate of the GDM group be much higher than that of the OVA group? Also, patients themselves may have tried to control their blood glucose levels, even if they were not advised to do so. These factors could have influenced the outcomes observed in the study.

8. Why do some women have a biometry scan at the time of the GCT and why do some not? Could it be because of additional risk factors? If so, this would bias the results and limit generalizability. If information could be provided regarding the characteristics of those that had a scan and those that dd not could help identify any baseline differences.

We greatly appreciate your insightful question. Since this was a retrospective study, there were no predetermined guidelines for subjects to undergo biometry scans at the time of the GCT. However, most obstetricians recommended having a scan between 24-28 weeks of gestation at the time of GCT. In our previous paper involving the same subjects, clinical characteristics between the two groups – with and without biometry - were analyzed and no statistically significant differences were found. So, we will add some text addressing this issue. 

“There were no significant differences in the clinical and biochemical characteristics of the groups with and without fetal biometric data at the time of GCT.” (Line 111-113)

9. I have to admit that I don’t exactly understand the creation of the GA-AC. I assume you converted the measured AC (mm) to the corresponding mean AC (mm) by gestational AC nomograms. First, giving an example would make it clearer. Second, I am not understanding why you would not just report percentiles for the AC instead.

We sincerely apologize for any confusion caused by the unclear description. We have now revised it as below. 

“The mean values were converted to each estimated GA (i.e., GA-BPD, GA-FL, and GA-AC) according to the Japanese fetal growth chart.” (Line 119-120 in the “Revised Manuscript with Track changes” and lines 116-118 in the “Manuscript” file)

Additional explanation: 

1) AC (mm) was measured 3 times, the average was calculated, and the average AC (mm) was converted to estimated gestational age according to the growth chart. So, GA-AC is the estimated gestational age of AC of the fetus. Each value at the upper part of Table 2 is the mean estimated gestational age for each fetal part in the target group.

2) Measuring AC by fetal ultrasound can be used as a reliable marker of fetal adiposity (doi: 10.2337/dc16-0160). In our study, we calculated FAOR as an indicator of FAO with the estimated GAs instead of the percentile values of fetal parts measured via ultrasonography. We tried to observe abdominal overgrowth relative to other parts of the fetus, such as the femur and head. FAO, defined as ≥90th percentile of FAOR, is a more sensitive index for the detection of fetal asymmetric overgrowth than the AC percentile (https://doi.org/10.1371/journal.pone.0225955, page 10). As shown in Supplementary Table 2, the prevalence of LGA at birth in the NGT and GDM groups was lower than the prevalence of FAO near term, which suggests that some infants have relative abdominal overgrowth but appropriate body weight for GA at birth. This finding is consistent with the report by Catalano et al. [12] indicating that infants of women with GDM have increased body fat despite having an average weight for GA. doi: 10.4093/dmj.2020.0078. 

10. How was the FAOR ≥90th percentile determined? Was the 90th centile determined internally from the cohort?

Yes, we determined the 90th percentile among the total subjects with fetal biometry internally. 

11. Please discuss the statistical power of the study for the outcome of FAO and APO. I doubt that the study was powered for significant adverse outcomes.

We completely agree with your concerns about whether this study has statistical power for adverse pregnancy outcomes because the case number of the OVA group with FAO was as small as 17. Therefore, we inserted the following sentence in the results section.

“For the OVA group with FAO, the number of cases was too small, despite the p-value being less than 0.05 in the proportion test for pregnancy outcome.” (Line 275-276 in the “Revised Manuscript with Track changes” and lines 273-274 in the “Manuscript” file)

12. I see no explanation regarding how the adverse perinatal outcomes were chosen as it seems quite arbitrary. I see no elements of severe maternal or neonatal morbidity which are of course the most significant. Were there cases of stillbirth after GDM screening?

We considered that the most important perinatal adverse outcome associated with maternal glucose intolerance is fetal abdominal overgrowth, which can result in neonatal LGA, macrosomia, and cesarean section. Among all subjects who underwent GDM screening, IUFD was observed in 6 cases, but none of the IUFD cases were associated with GDM or OVA. 

13. Just a general comment regarding the use of “adiposity”. You are not measuring adiposity by measuring the AC. This is a sum of fetal liver size, perhaps intraabdominal adiposity and subcutaneous fat. The liver size likely contributes the most to the AC measurement thus the term is inaccurate

We completely agree with you. Fetal visceral fat or abdominal subcutaneous fat was not measured. To overcome this problem, we used FAOR, calculated by dividing the estimated GA of AC with the estimated GA of FL or BPD, as an indicator of fetal obesity. 

Results:

14. Well controlled GDM should lead to near normalization of the fetal growth and this does not appear to be the case here. Could it be that your GDM population is suboptimally managed and this is the reason for some of these differences? Providing information on adequacy of glycemic control and/or further stratifying by method of control (medical\\lifestyle alone) would be helpful in interpreting the results.

According to our previous report (doi: 10.4093/dmj.2020.0078.), the average fasting blood glucose and 1hr postprandial blood glucose levels during follow-up after GDM diagnosis were 81.9±9.0mg/dL and 133.0±19.1mg/dL, respectively. Also, the average glycated albumin and HbA1c levels were 11.8±1.4% and 5.6±0.6% respectively. Infant birth weight did not show significant difference between NGT and GDM (3,197±421g vs. 3,226±510g), but gestational age at delivery was slightly higher in NGT than GDM subjects (39.0±1.5 vs. 38.4±1.8, p<0.05). However, the odds ratio for fetal abdominal obesity in GDM subjects was significantly higher than that in NGT subjects. Furthermore, the prevalence of LGA and

---

## [Decision Letter · Decision Letter 1]

21 May 2024

Fetal Abdominal Obesity in Women with One Value Abnormality on Diagnostic Test for Gestational Diabetes Mellitus

PONE-D-23-38854R1

Dear Dr. Kim,

We’re pleased to inform you that your manuscript has been judged scientifically suitable for publication and will be formally accepted for publication once it meets all outstanding technical requirements.

Kind regards,

Clive J. Petry, PhD

Academic Editor

PLOS ONE

Additional Editor Comments (optional):

Reviewers' comments:

Reviewer's Responses to Questions

**Comments to the Author**

1. If the authors have adequately addressed your comments raised in a previous round of review and you feel that this manuscript is now acceptable for publication, you may indicate that here to bypass the “Comments to the Author” section, enter your conflict of interest statement in the “Confidential to Editor” section, and submit your "Accept" recommendation.

Reviewer #1: All comments have been addressed

2. Is the manuscript technically sound, and do the data support the conclusions?

Reviewer #1: Yes

3. Has the statistical analysis been performed appropriately and rigorously? 

Reviewer #1: Yes

4. Have the authors made all data underlying the findings in their manuscript fully available?

Reviewer #1: Yes

5. Is the manuscript presented in an intelligible fashion and written in standard English?

Reviewer #1: Yes

6. Review Comments to the Author

Reviewer #1: thank you for allowing me to re-revise this manuscript. the authors have satisfactorily addressed the reviewer's comments.

7. PLOS authors have the option to publish the peer review history of their article (what does this mean?). If published, this will include your full peer review and any attached files.

Reviewer #1: No

---

## [Editor Report · Acceptance letter]

24 May 2024

PONE-D-23-38854R1 

PLOS ONE

Dear Dr. Kim, 

I'm pleased to inform you that your manuscript has been deemed suitable for publication in PLOS ONE. Congratulations! Your manuscript is now being handed over to our production team.

Kind regards, 

on behalf of

Dr. Clive J. Petry 

Academic Editor

PLOS ONE